# Phase-specific transcriptional patterns of the oomycete pathogen *Phytophthora sojae* unravel genes essential for asexual development and pathogenic processes

Min Qiu[1,2,3☯], Mengjun Tian[1☯], Saijiang Yong[1], Yaru Sun[1], Jingting Cao[1], Yaning Li[1], Xin Zhang[1], Chunhua Zhai[1], Wenwu Ye[1,2,3], Ming Wang[1,2,3]*, Yuanchao Wang[1,2,3]*

1 Department of Plant Pathology, Nanjing Agricultural University, Nanjing, Jiangsu, China, 2 The Key Laboratory of Plant Immunity, Nanjing Agricultural University, Nanjing, Jiangsu, China, 3 Key Laboratory of Soybean Disease and Pest Control (Ministry of Agriculture and Rural Affairs), Nanjing Agricultural University, Nanjing, Jiangsu, China

☯ These authors contributed equally to this work.
* mwang@njau.edu.cn (MW); wangyc@njau.edu.cn (YW)

**Data Availability Statement:** All relevant data are within the manuscript and its Supporting information files.

## Abstract

Oomycetes are filamentous microorganisms easily mistaken as fungi but vastly differ in physiology, biochemistry, and genetics. This commonly-held misconception lead to a reduced effectiveness by using conventional fungicides to control oomycetes, thus it demands the identification of novel functional genes as target for precisely design oomycetes-specific microbicide. The present study initially analyzed the available transcriptome data of the model oomycete pathogen, *Phytophthora sojae*, and constructed an expression matrix of 10,953 genes across the stages of asexual development and host infection. Hierarchical clustering, specificity, and diversity analyses revealed a more pronounced transcriptional plasticity during the stages of asexual development than that in host infection, which drew our attention by particularly focusing on transcripts in asexual development stage to eventually clustered them into 6 phase-specific expression modules. Three of which respectively possessing a serine/threonine phosphatase (PP2C) expressed during the mycelial and sporangium stages, a histidine kinase (HK) expressed during the zoospore and cyst stages, and a bZIP transcription factor (bZIP32) exclusive to the cyst germination stage were selected for down-stream functional validation. In this way, we demonstrated that PP2C, HK, and bZIP32 play significant roles in *P. sojae* asexual development and virulence. Thus, these findings provide a foundation for further gene functional annotation in oomycetes and crop disease management.

## Author summary

Oomycetes are fungi-like microorganisms with differences in biology and genetics. Using fungicides to control the pathogenic oomycetes has limited effectiveness. Therefore, a detailed understanding of the oomycetes is necessary for identifying potential candidates

**Funding:** This work was supported by the National Natural Science Foundation of China grants (31721004 to YW, 32100044 to MW, and 32100160 to MQ) (http://www.nsfc.gov.cn/), and the China Postdoctoral Science Foundation grant (2020M681642 to MQ) (https://jj.chinapostdoctor.org.cn/), and the Chinese Modern Agricultural Industry Technology System grant (CARS-004-PS14 to YW) (http://www.moa.gov.cn/). The funders had no role in study design, data collection and analysis, decision to publish, or preparation of the manuscript.

**Competing interests:** The authors have declared that no competing interests exist.

as fungicide targets. The present study investigated the transcriptional changes in a model oomycete pathogen, *Phytophthora sojae*, during asexual development and plant infection. The analysis identified various types of genes with differences in expression patterns during development and infection. Further analysis focused on the transcriptome dynamics during asexual stages which exhibited more remarkable plasticity. Detailed analysis of the genes revealed stage-specific responses of a few genes and specifically identified three genes with functional relevance in different developmental stages of *P. sojae*, including a serine/threonine phosphatase in mycelial and sporangium stages, a histidine kinase in zoospore and cyst stages, and a bZIP transcription factor in cyst germination. Also these three genes function in different participate in multiple layers of the pathogenicity process. Thus, the study provides a reference for further research on the candidate genes, which may be used to defeat the pathogenic oomycetes.

## Introduction

Oomycetes, currently classified as stramenopile eukaryotes, are unicellular protists that physically resemble filamentous fungi; however, they are phylogenetically distinct and have unique biological, genetic, and physiological features [1]. Among the oomycetes, the *Phytophthora* and *Pythium* species and the downy mildews, known for their devastating effects on crops, vegetables, forests, and natural ecosystems [2]. Although oomycete pathogens display fungus-like growth pattern, most physiological processes targeted by fungicides are absent or dispensable in oomycetes, leading to the limited effectiveness of chemicals against oomycete diseases. For example, oomycetes lack most enzymes for melanin biosynthesis [3], which are important enzymes targeted by fungicides [4]. More pronounced differences between oomycetes and fungi have been revealed at the cellular and molecular levels. For example, the cell wall of fungi is composed of chitin, while oomycetes have cellulose and β-glucan. Researchers have compared the biological phenotypes of oomycetes and fungi using the classical microbiological approaches [5]; however, little is known about the molecular features underlying their disease cycle due to the lack of an efficient gene editing technology. Therefore, studies should aim at understanding the molecular characteristics and mechanisms underlying the oomycete life cycle and identify effective targets to combat the diseases.

In oomycete pathogens, the motile zoospore is the primary agent of dispersal under moist or wet conditions; these zoospores are able to use both general (ethanol) and host-specific signals to swim toward the host roots [6,7]. Most oomycetes produce numerous multinucleate sporangia at the end of specialized hyphae that grow on the surface of plant tissues [6]. The cytoplasmic cleavage and membrane reorganization of mature sporangia leads to the release of uninucleate and biflagellated zoospores [8]. Zoospores usually travel a few centimeters through wet soil and even farther through rain or irrigation water [9]. Upon stimulation by host exudates, these active zoospores swim towards the host and encyst, immediately shed their flagella to form a cyst wall, and secrete glycoproteins to adhere to the host surface. Within a few hours, these cysts extend germ tubes that swell to breach the plant cuticle and epidermal cell wall [5,10] and lead to the spread of the disease from one host to another. Thus, plant infection primarily depends on the ability of oomycetes to release zoospores from sporangia. Moreover, for successful infection, the oomycete pathogen, especially the host-specific ones that persist poorly in a natural environment, should adjust and survive during the transition from one stage to another.

*Phytophthora sojae*, one of the model oomycetes, is a devastating soil-borne pathogen. It specifically infects soybean and causes seedling "damping off" and root rot. It is a good model for studying the molecular features of the oomycete life cycle because it can be easily cultured *in vitro*, and all stages can be induced without a plant host [11]. Moreover, the genome and transcriptome data of *P. sojae* are available [12,13]. The recently established clustered regularly interspaced short palindromic repeats (CRISPR)-mediated gene knockout system has also strengthened functional genomic research in *P. sojae* [14,15]. Over the past dozen years, potential genes involved in different life stages of *P. sojae* have been identified through gene silencing or knockout technologies [16–19]. Despite such advances in the oomycetes kingdom, many molecular mechanisms underlying their life cycles remain unknown, especially during the critical stages of disease spread. Recently, a combination of transcriptome and functional analyses has proved to be a valuable approach for gaining insight into pathogen asexual development and pathogenesis [20–22]. Therefore, we should aim to identify and characterize functional genes facilitating developmental using genome editing techniques and available data on oomycetes.

The present study performed large-scale transcriptome profiling across the asexual development and infection stages to analyze the dynamics of the *P. sojae* life cycle. We found remarkable transcriptional changes during asexual development and classified the genes into six modules based on expression patterns. Furthermore, the study identified and functionally characterized three candidate genes with stage-specific significance. Thus, the study's findings will provide new insight into the pathogenesis-associated genes of oomycetes.

## Results

### *Phytophthora sojae* exhibits greater transcriptional plasticity during asexual development than during host infection

A gene expression matrix was constructed using the transcriptional data (normalized gene expression levels) of *P. sojae* across asexual development and during host infection obtained by 3'-tag sequencing [13]. The five representative asexual stages (mycelia, sporangia, zoospore, cyst, and germinating cyst, hereafter referred to as MY, SP, ZO, CY and GC respectively) and five infectious stages with 1.5, 3, 6, 12 and 24 hours after *P. sojae* inoculation on soybean leaf (IF1.5h, IF3h, IF6h, IF12h and IF24h as abbreviations) are shown in Fig 1A. A total of 10,953 genes were retained in the matrix after filtering the ones with low abundance (total counts less than 10). The hierarchical clustering after Z-MAD normalization (relative median absolute deviation) representing variation coefficients was displayed in a 2D-heatmap space, illustrating a clear deviation at developmental and infectious level: infectious samples clustered in one clade, whilst developmental samples were dissociated with them (Fig 1B), suggesting a vast difference in transcriptional profiles at group layer between asexual development and host infection. Notably, for these five infectious samples, majority of genes being expressed in at least two of infectious stages shared gene identities, displaying an agglomeration in the central area. On the contrary, genes found in these samples during asexual stages preferentially expressed in only one or a few samples, rather than in all five, as a result, these highlighted cells in red (highly expressed genes) more sporadically located at different space of heatmap than that in infectious samples (Fig 1B). These prominent changes in gene expression inferred a greater transcriptional plasticity during the five stages of asexual development than that during the stages of infection.

To deeply capture a holistic picture for these transcriptomic profiles, transcriptomic specificity and diversity were estimated by introducing two information theory descriptors: Hj index for quantification of transcriptome diversity, and δj index for transcriptomic profile

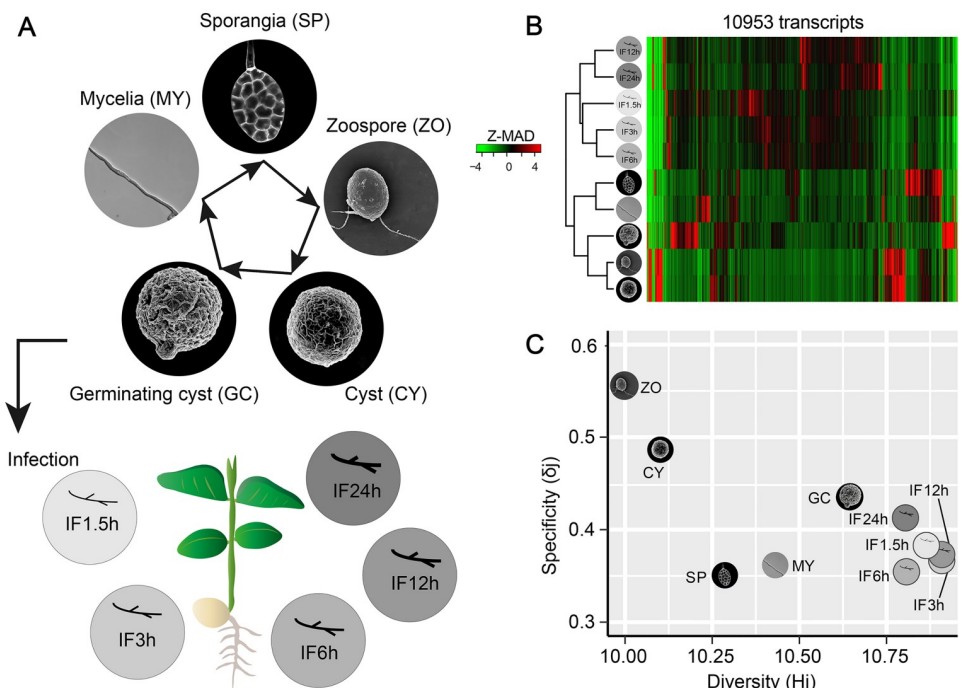

**Fig 1. Overview of transcriptome data set with host infection and asexual development in *P. sojae*.** (A) Schematic diagram of asexual development and infection stages. Asexual stages including Mycelia (MY), Sporangia (SP), Zoospores (ZO), Cyst (CY) and Germinating cysts (GC). Infection stages includes IF1.5h, IF3h, IF6h, IF12h, IF24h, which mean that samples were taken after 1.5, 3, 6, 12, 24 h inoculation onto susceptible soybean leaves. (B) Hierarchical clustering of expression matrix of transcripts in asexual and infection stages. The expression matrix was removed of transcripts with low expressions. (C) Correlation analysis of expression data from ten asexual and infection stages. x-coordinate: Hj index for quantification of transcriptome diversity; y-coordinate: δj index for transcriptomic profile specialization.

specialization. These two information descriptors were depicted in the 2D scatter space in Fig 1C with gauged values, which demonstrated all samples of infection stages (IF1.5h, IF3h, IF6h, IF12h and IF24h) grouped in a close proximity in the bottom right corner (Fig 1C), suggesting a higher diversity but lower specificity of the transcriptome. In other words, genes displayed similar expression patterns at the various stages of infection, indicating low transcriptional plasticity. In contrast to the infection stages, the five asexual stages were scattered more sporadically in the plot (Fig 1C), indicating larger differences in the diversity and specificity of the transcriptome. In the plot, the zoospore stage (ZO) and the cyst stage (CY) exhibited high specificity but low diversity. They were close to each other (Fig 1C), suggesting that both stages had a small set of genes with drastically expression. The mycelial (MY) and the sporangia (SP) phases, which were close to each other, had moderate diversity and low specificity (Fig 1C), indicating genes expression shared a relative narrow range. The germinating cyst (GC) appeared separation from the other four asexual stages; it was closer to the infection stages but displayed higher specificity (Fig 1C), indicating an increased number of genes with preferentially expression in GC stage.

Taken together, both hierarchical clustering and information theory descriptors concertedly retrieved a highly dynamic transcriptional profile of *P. sojae* during the asexual stages, which referred as we describe as transcriptional plasticity. Therefore, we further solely focused on the stages within asexual development for downstream expression pattern recognition and functional validation.

## Gene expression modules of *Phytophthora sojae* during asexual development

Next, we analyzed the transcriptional profiles across the five asexual stages to explore gene expression patterns. First, we removed genes with no changes in abundance across the stages and obtained a new expression matrix based on remained 7028 genes. Then, a previously reported Short Time-series Expression Miner (STEM) analysis was performed [23] to determine the temporal expression (transcript abundance) patterns. The expression matrix of 7028 genes was eventually resolved into six modules with uniqueness in expression patterns (Fig 2A–2F). In the radar charts of each module, the five stages (MY, SP, ZO, CY and GC) are represented by five circles along the axis. The values along the radar axis represents the relative transcript abundance of these clustered genes, and the biological functions terms of gene ontology (GO) significantly enriched within each module are displayed next to each radar chart (Fig 2A–2F). The M1 module had 1553 genes. In this radar chart, a peak value was observed along the GC axis (Fig 2A), indicating genes were dominantly expressed during the

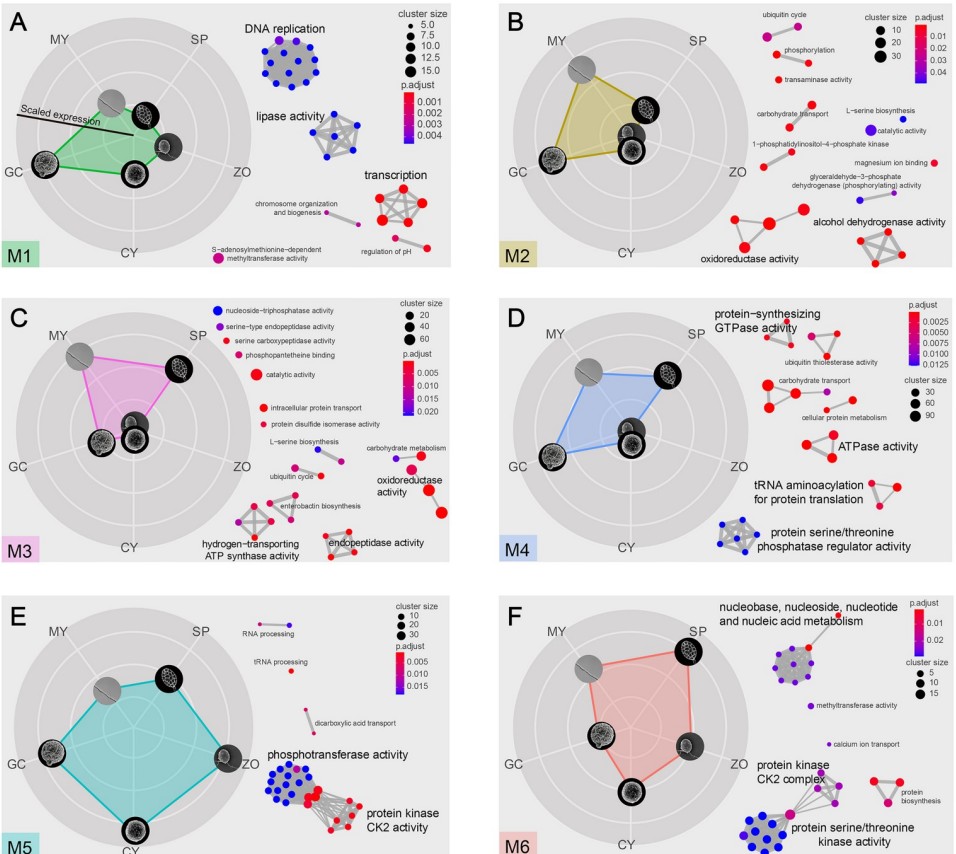

**Fig 2. Transcription patterns and functional enrichment of clustered genes.** Normalized expressions of all genes were clustered into six models using STEM. Significant modules are shown in different colors. (A-F) Left: Rader maps display the average expression levels in each expression models. In each module, the five rounded pictures represent the five asexual stages (MY: mycelia; SP: sporangia; ZO: zoospores; CY: Cyst; GC: germinating cysts), and the distance from the center of the map represents the overall mRNA levels of each stage compared with the mean of the five stages. Right: the right side illustrates enriched functional clusters in each module. GOs (Gene Ontology) enrichment analysis of included genes in each expression models. GOs are represented by circles sized according to the number of expressed genes per GO. GOs significantly enriched in genes in a chi-squared test (adjusted *p < 0.01* after Bonferroni correction for multiple testing) are colored according to their enrichment score.

GC stage and represented significant enrichment of three primary functions, including DNA replication, transcription, and lipase activity (Fig 2A), implying their essential roles during sgerminating cyst stage by regulating nucleotide related activities under the frame of central dogma.

M1 showed high expression mainly at one stage, while those of the remaining five modules showed expression at multiple stages. M3, M4, and M6 genes showed high expression at the mycelial and sporangium stages (Fig 2C, 2D and 2F). M3 with 1539 genes showed higher expression levels during the MY and SP stages than in any other stage. In contrast, M4 and M6, with 2391 and 501 genes, respectively, exhibited high expression not only during the MY and SP stages but also in additional one (GC in M4) or two other stages (GC and CY in M6). The modules M6 and M4 included many regulatory genes encoding protein kinases and serine/threonine phosphatases respectively as key regulators mediating signaling transduction, which may act as potential candidates for further functional studies.

The M5 module had 605 genes, of which the majority showed an expression opposite to that observed for the three modules discussed above. The M5 genes showed higher expression during the zoospore and cyst stages and moderate expression during the germinating cyst stage (Fig 2E). Interestingly, this module contains much larger portion of ZO- and CY-specific genes relative to the others and GO enrichment analysis of these 605 genes identified a markedly pronounced phosphotransferase activity as the significantly enriched function (Fig 2E), suggesting a certain specialized function acting during zoospore and cyst stage, or during transition from zoospore to cyst.

As to the last module, M2 had the minimum number of genes (439), with high expression levels during the mycelial and germinating cyst stages (Fig 2B). Genes of this module showed significant enrichment in dehydrogenase and oxidoreductase activities and were associated with catabolism, indicating a set of metabolic reactions take place within these stages. To this end, we found an association of particular genes within each module forges a landscape of specific dynamics in biological functions, that infers specialized physiological activities functioning within given stages or during transition from one stage to another.

## Modules with high gene expression during the mycelial and sporangium stages

We then focused on each module to identify the unique molecular functions of the genes during the transition of *P. sojae* from one stage to another. The modules M3, M4, and M6 included mRNAs with high abundance at the mycelium and sporangium stages; M3 and M4 were less abundant at the zoospore and cyst stages, while M6 was moderately abundant; M4 was abundant at the germinating cyst stage, but M3 and M6 were less abundant (Fig 2C, 2D and 2F). These three modules (M3, M4, and M6) had numerous genes (4431; 63% of the total), of which M4 was the largest, with 2391 genes, followed by M3 (1539) and M6 (501).

Detailed analysis showed that three modules had genes with high expression only during the mycelium and sporangium stages (M3) or with additional one or two stages (M4 and M6), suggesting exclusive and shared functions. For example, genes in M3 participated in biological processes unique to the mycelial and sporangium stages, such as sporangium maturation. We also found that the genes encoding proteins with hydrolysis-related activities appeared in M3 (Fig 2C) but not in other modules. This observation indicates that hydrolysis-associated cellular function may be exclusive to the mycelium and sporangium stages. The genes of M4 and M6 were expressed in various stages (including the mycelium and sporangium stages; Fig 2D and 2F) and suggested their shared functions. We noted that the M4 and M6 modules contained several genes in the serine/threonine phosphatase and serine/threonine kinase families.

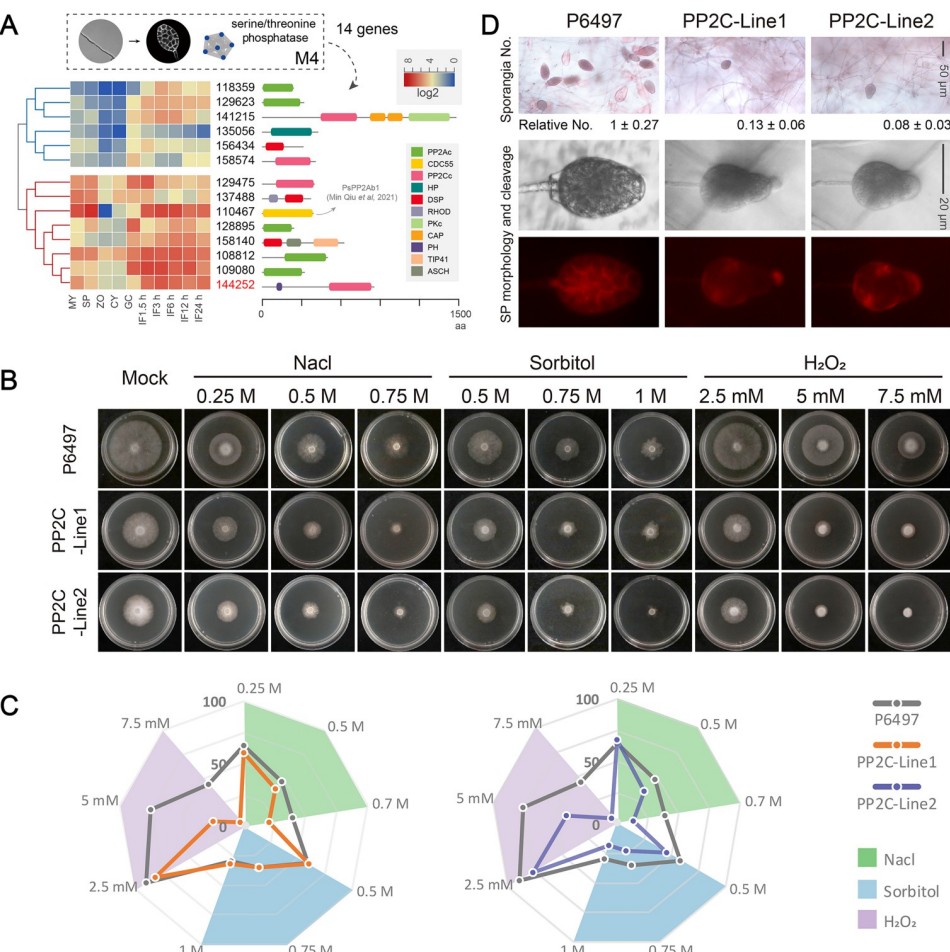

**Fig 3. Serine/threonine phosphatase (PP2C) required for sporangia formation and mycelial growth in *P. sojae*.**
(A) Hierarchical clustering of transcription levels of the genes from serine/threonine phosphatase cluster. Domain structure of each gene were displayed on the right panel. (B) Growth characteristics of the wild-type (P6497) and PP2C-knockout mutants (PP2C-Line1 and PP2C-Line2) on V8 agar medium without (Mock) or with different concentrations of NaCl, Sorbitol or $H_2O_2$. (C) Rader maps display the relative colony diameters after 4 days of growth in different stresses. Colony diameters were measured in each independent biological experiment after 4 days of growth. And Relative colony diameters were calculated for each treatment relative to growth on V8 agar medium only. Different stress treatments were colored with pale green (Nacl), thistle ($H_2O_2$), and sky blue (Sorbitol). (D) Upper panel: The numbers of sporangia in wild-type (P6497) and PP2C-knockout mutants (PP2C-Line1 and PP2C-Line2). The relative numbers of sporangia were labeled under the pictures. Lower panel: The sporangia morphology and cytoplasm cleavage within sporangia were observed in the wild-type (P6497) and PP2C-knockout mutants (PP2C-Line1 and PP2C-Line2).

Given the importance of such proteins in signaling transduction, we drew intensive attention on this part of genes.

In some cases, genes associated with the same function exhibited differences across stages in the mRNA abundance levels, probably due to the differences in the importance of these genes in molecular functions. Then, to address this issue, we performed statistical clustering of mRNAs from all stages and visual inspection of protein domains to categorize the genes into subclasses. We classified the genes from the serine/threonine phosphatase cluster into two groups, one that was relatively less expressed and the other with consistently high expression at all stages of infection (Fig 3A). We also identified multiple phosphatase domains, such as

PP2A, CDC55, PP2C, HP, and DSP, in these serine/threonine phosphatase-encoding genes (Fig 3A).

## Protein phosphatase 2C is necessary for mycelial growth and sporangia production

Among the serine/threonine phosphatases, a putative protein phosphatase 2C, Ps144252 (PP2C), was identified with an additional PH domain; it showed consistently high expression during the asexual and infection stages. Then, to verify the biological function of PP2C in *P. sojae*, two independent knockout lines of PP2C (PP2C-Line1 and PP2C-Line2; S1A–S1C Fig) were generated using CRISPR/Cas9 mediated technique. After four days of growth on the V8 agar medium, the colonies of the PP2C knockout mutants appeared smaller than the control (P6497) (Fig 3B). However, the mutants in the presence of NaCl or sorbitol at different concentrations grew without significant difference relative to their growth at the V8 agar medium (Mock) (Fig 3B and 3C). Hydrogen peroxide ($H_2O_2$) at different concentrations reduced the growth of the PP2C knockout mutants compared to the mock medium (Fig 3B and 3C). In V8 agar medium, the wild-type *P. sojae* strain P6497 formed abundant sporangia, while PP2C knockout mutants (PP2C-Line1 and PP2C-Line2) exhibited a significant reduction (87%–92%) in sporangium number (Fig 3D). In oomycetes, zoospore release requires the cleavage of the sporangial cytoplasm by the membrane network [7]. Further observation showed that the PP2C knockout mutants produced smaller sporangia with undifferentiated cytoplasm (Fig 3D). These results indicated that *P. sojae* PP2C is required for sporangia production and sporangial cytoplasm cleavage, and the mutants had defects in zoospore release.

## Modules with genes expressed predominantly at the zoospore and cyst stages

After the mycelial and sporangium stages, we focused on the zoospore and cyst stages, which displayed similar transcriptome profiles (Fig 1B and 1C). The fifth module (M5) had a few gene clusters with peak expression during the zoospore and cyst stages, moderate expression in germinating cysts, and less expression during the mycelial and sporangium stages. Notably, this module had genes with kinase activity, including a large cluster with phosphotransferase activity (Fig 2E); these phosphotransferases are assumed to act similarly to kinases under certain conditions. Detailed analysis of this module showed 36 genes encoding proteins with phosphotransferase activity, which were divided into two groups based on transcriptional abundance (Fig 4A). Among the 36 genes, 24 were relatively less expressed in infection stages, and the remaining 12 were highly expressed during the asexual development and infection stages (Fig 4A). Most of these highly expressed genes encoding serine/threonine kinases that phosphorylate amino acid residues (8 of 12, STKc); the remaining encoded a phosphagen kinase (Ps132428), a thymidine kinase (TK, Ps110301), a phosphatidylinositol phosphate kinase (PIPK, Ps132153), and a histidine kinase (Ps130388) (Fig 4A).

Several studies have demonstrated the roles of histidine kinases in sensing environmental signals [24,25] and regulating sexual development and virulence [26,27]. The putative histidine kinase (Ps130388) indicated the importance of signal recognition and regulation in *P. sojae* during development. Therefore, we further investigated the histidine kinase of this module.

## A histidine kinase is necessary during the zoospore and cyst stages

The M5 had one histidine kinase (Ps130388, HK), which was highly regulated during the asexual and infection stages (Fig 4A). This HK had multiple PAS (Per Arnt Sim) domains, a TorS

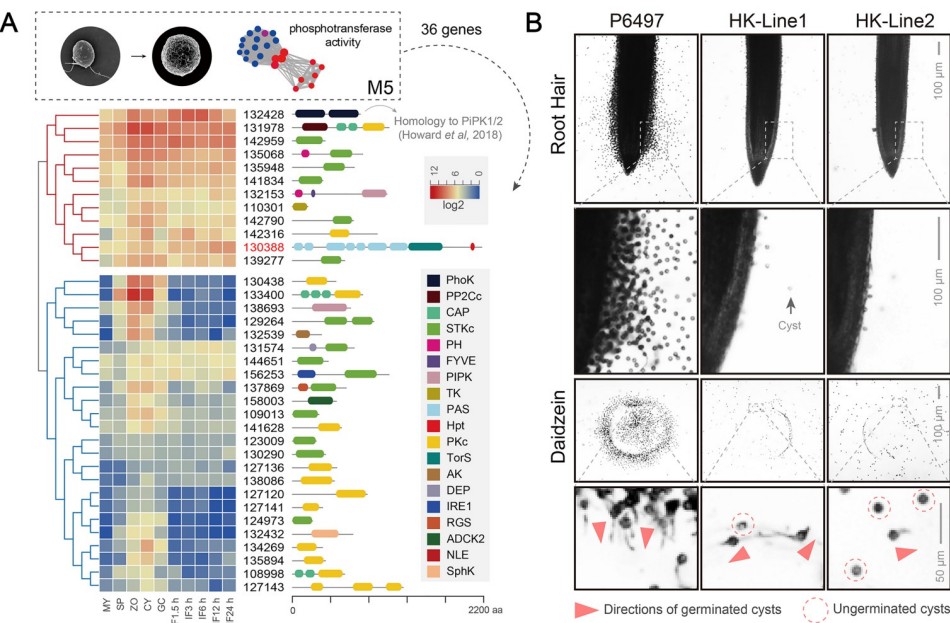

**Fig 4. A histidine kinase (HK) with phosphotransferase activity necessary for zoospore chemotaxis in *P. sojae*.** (A)
On the left shows that hierarchical clustering of the genes with phosphotransferase activity from fifth module in Fig 2.
And on the right show the corresponding domain structures. (B) Analyses of zoospore chemotaxis to root hairs and
agar blocks with daidzein. Root tips or agar blocks contains daidzein were placed in glass slides containing equal
amounts of zoospore suspensions from the wild-type (P6497) and HK-knockout mutants (HK-Line1 and HK-Line2).
Photos were taken after 5 min in root hairs and 20 min in agar blocks, while most of the zoospores got encysted. The
directions of germinated cysts were indicated with red triangles. The Germinated cysts were marked with dotted
circles. All experiments were repeated three times with similar results.

domain, and a histidine-containing phosphotransferase (HPt) domain. Studies have reported
that the TorS domain is a putative regulatory complex of the bacterial two-component system,
with histidine kinase-like ATPases (SM000387), his kinase A (phosphoacceptor) domain
(SM000388), and cheY-homologous receiver domain (SM000448) [28]. Therefore, to verify
the biological function of the HK in *P. sojae*, CRISPR/Cas9-mediated gene editing was per-
formed. Due to the low efficiency of HDR (Homology Directed Repair) events in long
stretches of DNA (full-length *HK* ORF is 6486 bp), a non-homologous end-joining (NHEJ)-
mediated CRISPR/Cas9 gene editing was performed to generate a direct deletion in the gene.
This way, two independent knockout lines were obtained with large deletions in HK (S1D and
S1E Fig, HK-Line1 and HK-Line2).

Next, we tested the zoospore chemotaxis of the HK knockout lines, using the wild-type
strain (P6497) as a control. As shown in Fig 4B (Root hair), the wild-type zoospores aggregated
to the soybean root hairs within 10 min. Meanwhile, in contrast zoospores of the HK knockout
mutants (HK-Line1 and HK-Line2) were not chemoattracted to root hairs. Zoospores of the
wild-type strain formed cysts on and adjacent to this region of the root. While only a few cysts
were observed in this region of the root in the knockout mutant strains (Fig 4B, Root Hair).
Subsequently, to explore the role of HK in zoospore chemotaxis, soybean root material was
replaced with an agarose plug containing the isoflavone daidzein, a component of the soybean
root exudate that attracts zoospores of *P. sojae* but not of other species [29]. Zoospores of the
wild-type strain swam rapidly to the plug and began to germinate within 20 min (Fig 4B, daid-
zein). In contrast, zoospores of the HK knockout mutants (HK-Line1 and HK-Line2) did not

aggregate to the agarose plugs (Fig 4B). In wild-type (P6497), closer examination revealed that the cysts germinated towards daidzein-containing agarose plugs shown in the bottom panel (Fig 4B). However, HK knockout mutants had fewer cysts, either ungerminated or germinated in irregular directions. These observations confirmed that HK expression is necessary for *P. sojae* zoospore chemotaxis.

## Module with the highest gene expression during cyst germination

Among the five asexual stages analyzed in this study, the germinating cyst stage appeared distinct from the other four stages (MY, SP, ZO and CY), exhibiting GC-specific expression pattern of many genes (Fig 1B and 1C). The first module (M1), with 1553 genes, had clusters specifically and highly expressed during cyst germination (Fig 2A). These gene clusters that significantly enriched DNA replication, lipase activity, and transcription did not exist in other modules (Fig 2A), which confirmed the uniqueness of the cyst germination stage. We found that the cluster in M1 with transcriptional activity had 32 genes, classified into three groups based on the expression levels: low expression (7 genes), moderate expression (12 genes), and high expression (13 genes) genes (Fig 5A). Various types of transcriptional activities were identified for these three classes, including RNA polymerase (RNA_pol), transcription elongation factor (TEF), transcription initiation factor (TIF), histone acetylation or methylation (HAT, SIN), and transcription factor (CBFB, HOX, HSF, PHD, and bZIP) activities. Among the consistently and highly expressed candidate genes, more than half encoded RNA polymerases and the remaining transcription factors (Fig 5A). Ps132948 (bZIP32), a putative bZIP transcription factor with an additional PAS domain, was highly expressed during cyst germination and the infection stages (Fig 5A). Studies have associated this transcription factor with cyst germination in *P. sojae* and *Peronophythora litchi* [30,31].

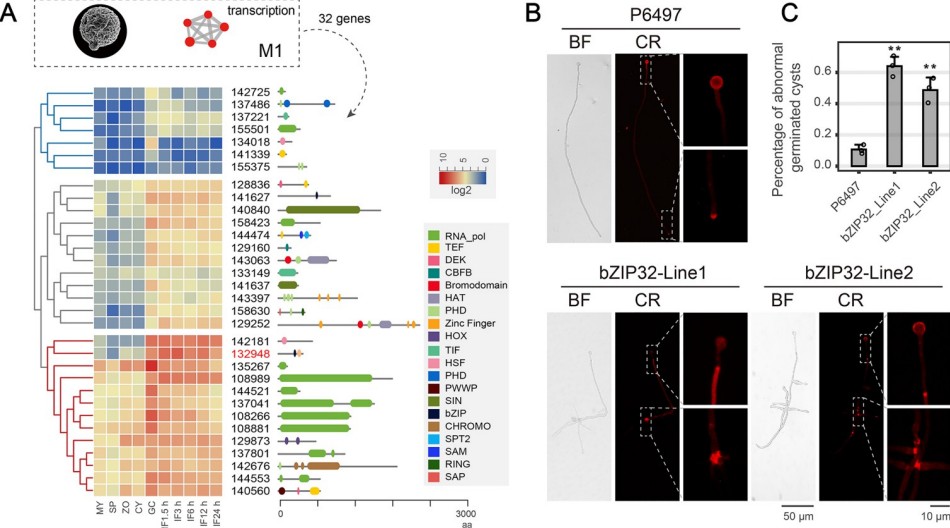

**Fig 5. A bZIP transcription factor (bZIP32) required for cyst germination in *P. sojae*.** (A) Hierarchical clustering of mRNA levels of genes involves in transcription activity in the first Module. And also the domain structures of included genes were displayed on the right panel. (B) Observation of germinated tubes at 8 h after zoospores encysted. Cysts from wild-type (WT) and bZIP32 silenced mutants (bZIP32-Line1 and bZIP32-Line2) were incubated at 25°C for 8 h and photographed. Bar, 50 μm. (C) The morphologies of germinated cysts were observed under a microscope, and the ratio of abnormal germinated cysts to the total number of cysts was calculated. All experiments were repeated three times with similar results. **Significant difference at *p < 0.01*.

## A bZIP transcription factor indispensable for cyst germination

To further investigate the role of Ps132948 (bZIP32) in GC stage, we used previously generated *bZIP32*-silenced mutants (bZIP32-Line1 and bZIP32-Line2) with reduced *bZIP32* transcript levels (>50%) (S1G Fig) [31]. In 50% (v/v) V8 liquid medium, 90% of the wild-type germ tubes maintained a tapering tip and a constant diameter (Fig 5B). However, approximately 50% of germ tubes of the *bZIP32*-silenced mutants (bZIP32-Line1 and bZIP32-Line2) displayed branches after germination (Fig 5B and 5C), indicating an early polarized growth. In addition, P6497, bZIP32-Line1, and bZIP32-Line2 exhibited 10%, 64%, and 48% abnormally germinated cysts, respectively (Fig 5C).

Furthermore, to explore the cause of abnormal germ tube formation, we analyzed the cell wall components, such as β-1,3-polysaccharides, β-1,6-polysaccharides, and cellulose [32,33], during cyst germination using Congo Red staining. After 5 min of staining, red fluorescence was detected in the circular cysts or the germ tube tips of the wild-type strain. In contrast, the fluorescence was evident only at the branching sites and not at the germ tube ends in the *bZIP32*-silenced mutants (Fig 5B). This observation indicates that regular polysaccharide transport contributes to the polarized growth of germ tubes.

## Asexual development-linked genes participate at multiple levels of pathogenicity

To further determine whether the role of these asexual development-related genes in host infection, we performed infection assays using the PP2C and HK knockout mutants, *bZIP32*-silenced mutants and wild-type control. Because of the PP2C mutants had defects in zoospore release, the assay used mycelial plugs from the wild-type strain (WT-MY) and the PP2C knockout mutants (PP2C-Line1 and PP2C-Line2). Soybean seedlings inoculated with wild-type plugs produced typical water-soaked spots around the disease lesions. In contrast, seedlings inoculated with the PP2C-Line1 and PP2C-Line2 plugs showed significantly smaller lesions (Fig 6A and 6B). The relative *P. sojae* biomass in the infected soybean seedlings was 80% lower for the PP2C knockout mutants than in the wild-type control (Fig 6C). Similarly, soybean seedlings inoculated with the zoospores from HK knockout mutants and bZIP32-silenced mutants developed only small lesions at the site of inoculation. In contrast, the seedlings inoculated with the wild-type zoospores produced typical water-soaked lesions (Fig 6A and 6B). The relative *P. sojae* biomass in soybean seedlings infected with HK-Line1, HK-Line2, bZIP32-Line1, and bZIP32-Line2 was significantly lower than that infected with the wild-type strain (Fig 6C).

To further disentangle the reason for the reduced pathogenicity of these mutants, epidermal cells from the seedling inoculation sites were observed after DAB staining. At 12 h post-inoculation (hpi), plenty of zoospores from the wild-type P6497 colonized the epidermal cells, and the invasive hyphae expanded into adjacent epidermal cells. The PP2C knockout mutants (PP2C-Line1 and PP2C-Line2) and *bZIP32*-silenced mutants (bZIP32-Line1 and bZIP32-Line2) resulted in obvious plant cell death around the infection sites (Fig 6D). The HK knockout mutants (HK-Line1 and HK-Line2) showed no cell death but fewer zoospores in the epidermal cells. Thus, all mutants displayed significantly reduced infectious ostioles (Fig 6E). At 24 hpi, P6497 had abundant infectious hyphae with tapering tips that extended into neighboring cells (Fig 6D). In contrast, the PP2C knockout mutants (PP2C-Line1 and PP2C-Line2) and *bZIP32*-silenced mutants (bZIP32-Line1 and bZIP32-Line2) induced severe cell necrosis in the plant cells (Fig 6D and 6F). In HK knockout mutants, no obvious cell death was found at 24 hpi relative to 12 hpi, and the abundance and length of invasive hyphae were markedly less (Fig 6D), which reduced infection. These observations suggest that PP2C, bZIP32, and HK play significant roles in *P. sojae* virulence through various strategies.

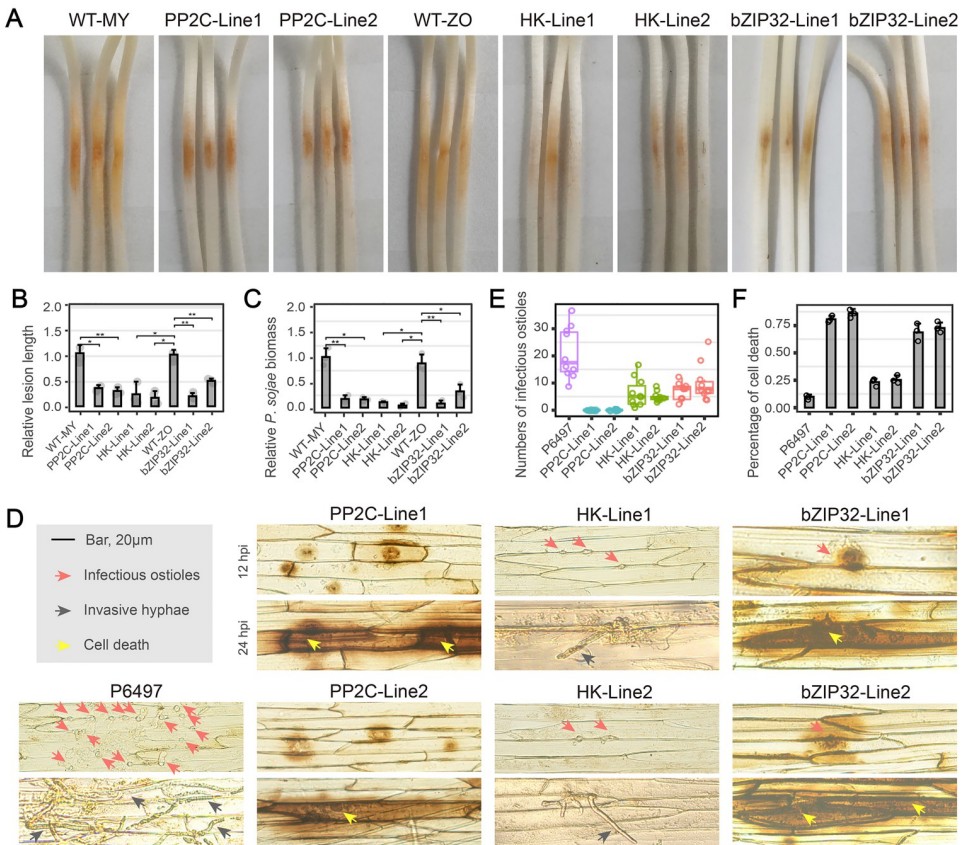

**Fig 6. Asexual development-linked genes function in virulence.** (A) Virulence assays of mycelium plugs from wild-type (WT-MY), PP2C-knockout mutants (PP2C-Line1 and PP2C-Line2) or of zoospores from the wild-type (WT-ZO), HK-knockout mutants (HK-Line1 and HK-Line2), and bZIP32-knockout mutants (bZIP32-Line1 and bZIP32-Line2). Etiolated hypocotyls were inoculated with same size of mycelium plugs or equal numbers of zoospores (100 zoospores/5–10 μL) and incubated at 25°C in the dark. (B and C) Relative lesion length (B) and pathogen biomass (C) in inoculated hypocotyls expressed as the ratio of the amounts of *Phytophthora sojae* DNA to soybean DNA detected at 48 hpi, with the *P. sojae*/soybean ratio set at 1. All experiments were repeated three times with similar results. **Significant difference at *p < 0.01*. *Significant difference at *p < 0.05*. (D) Microscopic observations of infectious ostioles and invasive hyphae in epidermis of soybean hypocotyls at 12 hpi and 24 hpi. DAB staining was performed on the epidermis of seedling hypocotyls. Red arrowheads, infectious ostioles; gray arrowheads, invasive hyphae; bright yellow arrowheads, cell death. Bar, 20 μm. (E) Numbers of infectious ostioles in epidermal cells of soybean hypocotyls at 12 hpi. (F) Percentage of cell death response of inoculated soybean hypocotyls of wild-type (P6497), PP2C-knockout mutants (PP2C-Line1 and PP2C-Line2) and HK-knockout mutants (HK-Line1 and HK-Line2), and bZIP32-knockout mutants (bZIP32-Line1 and bZIP32-Line2) at 24 hpi. In each sample 50 invaded epidermal cells were examined and the experiments were repeated three times.

In summary analysis of transcriptional profiling across the asexual developmental stages of *P. sojae*. Has enabled us to functionally characterize several key developmental genes facilitating the shift of gene expression form one stage to the next (Fig 7). These key genes include a serine/threonine phosphatase (mycelial to sporangium stages), a histidine kinase (zoospore to cyst stages), and a bZIP transcription factor (cyst germination) (Fig 7). And ultimately these key genes participate in *P. sojae* pathogenesis through various strategies.

## Discussion

Our analysis provides insights into the transcriptional diversity in *P. sojae* and other oomycetes during asexual development. Based on the expression data of *P. sojae* genes during asexual

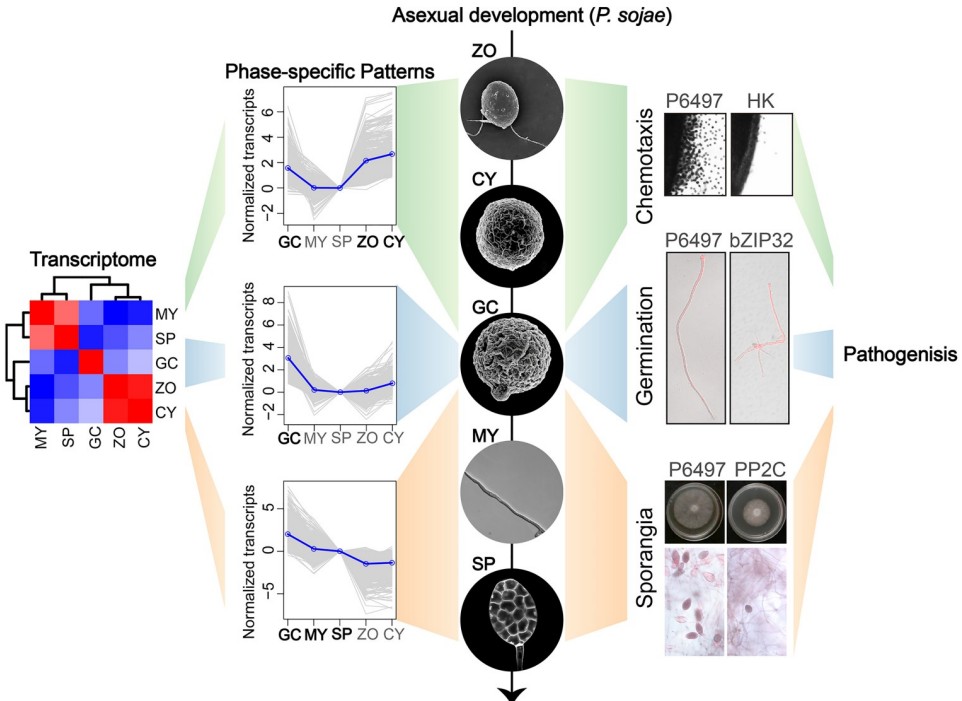

**Fig 7. Transcriptional profiling identifies novel functional genes of *P. sojae* associated with asexual development and pathogenesis.** This model proposes a possibility to study phase-specific genes that may play essential roles specifically in one or a few stages in *P. sojae* lifestyle by means of a combination of an explicit data-analysis pipeline and a molecular genome editing tool. By performing hierarchical clustering, specification, and diversification analyses, we focused on the stages of asexual development that are more remarkable in transcriptional plasticity, leading to transcriptome data being resolved into distinct phase-specific patterns. Next taking good advantages of genome editing in *P. sojae*, knockout mutants were generated for three candidate stage-specific genes and then found these genes function at specific asexual developmental stages, associated with their stage-specific expressions. These include a histidine kinase functioning in zoospore and cyst stages, a transcription factor in cyst germination, and a serine/threonine phosphatase in mycelia to sporangia stages. These novel functional genes participate in multiple layers of the pathogenicity process, which providing potential targets for precisely controlling diseases.

developmental and host infection [13], an expression matrix including 10,953 transcripts was obtained. We found that the asexual stages exhibited greater diversity and specificity in mRNA levels than during the process of infection, reflecting a striking degree of transcriptome remodeling during asexual development, which is related to the dramatic structural and physiological changes [7,8].

Further clustering analysis based on the expression of genes during the five asexual stages of *P. sojae* identified six modules, among which half (63% of the total) had genes co-expressed in the mycelial and sporangium stages (Fig 2C, 2D and 2F). The high correlation between the transcriptome patterns of these two stages may be because the sporangia are inseparable from hyphae in *P. sojae*. Numerous genes were also identified during these stages, probably due to the tremendous morphological changes occurring during sporangia formation. *Phytophthora* sporangia remain hydrated and metabolically active; they require specialized mechanisms to maintain viability during cytoplasm cleavage and store energy for zoospore release [6,34]. The genes associated with DNA helicase and serine/threonine phosphatase and the high mRNA levels of serine/threonine kinase during the mycelial and sporangium stages supported this assumption (Fig 2). The activity of enzymes associated with robust DNA metabolism, genome stability [35,36], cell growth, metabolism, and division [37,38] indicate the

tremendous changes around the sporangium stage. More importantly, the serine/threonine phosphatases analyzed in this study (gene ID 110467 in Fig 3A) have been found indispensable for mycelial growth and sporangia formation [39] in addition to PP2C (Fig 3B and 3D). However, HK and bZIP32 did not function in mycelia and sporangia development (S2 Fig), in contrast to PP2C essential for mycelial growth and sporangia formation, confirming the phase-specific expression and function of genes during asexual development.

Approximately 22% of the genes (Fig 2A; M1, 1553) were specific to the cyst germination stage. Stage-specific high abundance of the genes indicates transcriptome changes during the transition from one stage to another, probably because many transcripts are required to generate specialized cellular structures before infection or establish a compatible environment for successful host penetration. The significant enrichment of DNA replication, transcription, and lipase activity by the M1 genes approves this argument (Fig 2A). Our further investigation indicated that a bZIP transcription factor was essential for successful cyst germination (Fig 7), consistent with the observations in *P. sojae* and *P. litchi* [30,31].

Genes at the zoospore and cyst stages significantly represented phosphotransferase activity (Fig 2). This cluster included a phosphagen kinase that stores energy in the taurocyamine in animal cells [40], implying energy requirement during zoospore and cyst stages. Previous work in *P. infestans* demonstrated the functions of phosphagen kinase during the zoospore stage, such as integrating transcriptional regulation, metabolic dynamics, and protein retargeting [41]. In addition to the energy supplied by the phosphagen kinases, a putative histidine kinase implied signal recognition in the zoospore and cyst stages. In bacteria, histidine kinases sense environmental stimuli and regulate physiological changes; these kinases, together with response regulators, form the two-component signal transduction pathway [24,25]. Many fungi have hybrid HKs (HHKs), in which the HK and RR domains coexist in a single polypeptide. These HHKs regulate various processes, such as environmental perception, cell cycle control, and morphological change from nonpathogenic to pathogenic forms [26,27,42]. Interestingly, the histidine kinase in our study was also a hybrid HK, containing HK integrated with RR domains (Fig 4A). This hybrid HK regulated zoospore chemotaxis and subsequent cyst germination (Fig 4B). Previous studies identified that histidine kinase plays an important role during *P. infestans* infection [43]. Our experiments using knockout mutants support these earlier findings. However, there is a lack of in-depth knowledge of the two-component systems in oomycetes, especially in regulating downstream cellular signaling.

The second module had the lowest number of genes (439 genes; 6%) but with higher expression levels in the mycelia and germinating cysts (Fig 2B, M2). These genes might be involved in regulating metabolism and nutrient uptake. Many genes from M2 were associated with dehydrogenase activity, oxidoreductase activity, metabolism, and sugar transport. These facts reflect their significance in preparing germinated cysts to assimilate external nutrients, similar to hyphae. However, this contrasts the modules significantly enriched at the mycelial and other stages, when many compounds are transported from the plant and used as metabolic substrates. The evidence to support this may be that many M2 genes encode proteins involved in dehydrogenase activity, glycolysis, and sugar transportation (Fig 2B).

Previous studies reported the transcriptomic dynamics in oomycete pathogens such as *P. infestans*, *P. sojae*, *Phytophthora parasitica* [13,44,45]. However, most of them are focused on data analysis and further functional verifications were largely limited due to the lack of efficient gene editing systems. Thus, the present study performed large-scale transcriptional profiling during the asexual stages, and then took good advantages of genome editing in *P. sojae* to verify key genes associated with asexual development. Future studies should compare various oomycete species with similar or distinct asexual cycles and identify functional clusters

conserved across the oomycete kingdom; this study will help define the molecular features of oomycete species.

# Materials and methods

## Transcriptome analysis

The transcript analysis of *P. sojae* asexual and infection stages was based on RNA-seq (Ye *et al.*, 2011). Representative values for transcripts abundances are FPKM (Fragments Per Kilobase Million). At first, the FPKM data matrix was log2 transformed. Then, with the sporangia (SP) as the initial point, this transformed expression matrix of 10953 genes was normalized to 0 on the basis of algorism embedded in short Time-series Expression Miner (STEM) [23], while the transcript abundance of rest four points was scaled relative to SP. During this process, genes with low expression were cut off when their total counts below the threshold (10 in this study). As a result, 7028 genes exhibiting phase-specific (only for asexual stages) expression pattern were resolved into 6 modules. Regarding to statistics in pattern recognition, STEM was conducted followed default settings. In standard STEM pipeline, scaled method "Normalize data" was conducted transforming the vector to $(0, v1 − v0, v2 − v0, . . ., vn − v0)$, in which v0 is SP.

The gene expression data from three representative gene families, including transcription, serine/threonine phosphatase, and phosphotransferase activity, were clustered by HeatMap from TBtools [46]. First of all, the expression matrix of candidate genes was normalized on the basis of a global standardization. Then a Euclidean distance was estimated to build associations between the data points and their features. In this step we can see which points are similar or different from each other. Finally, we selected the complete-linkage as the cluster method and a dendrogram containing similarity and clustering information was generated (Figs 3A, 4A and 5A).

## Strains and culture conditions

The genome-sequenced *P. sojae* strain P6497 (Race 2, Tyler *et al.*, 2006, doi.org/10.1126/science.1128796) [12], generously provided by Professor Brett Tyler (Department of Botany and Plant Pathology, Oregon State University, USA), was used as the wild-type strain. The wild-type strain and all transgenic lines were routinely grown on 10% (vol/vol) V8 agar medium at 25˚C in the dark.

## CRISPR/Cas9-mediated gene editing techniques

Two sgRNAs (S2 Table, PP2C-sgRNA1 and PP2C-sgRNA2) were designed for targeting PP2C. Four sgRNAs, HK-sgRNA1, HK-sgRNA2, HK-sgRNA3, and HK-sgRNA4 (S2 Table), were designed for targeting HK. The design approaches were proceeded as previously described (Fang and Tyler, 2016).

For PP2C mutants, CRISPR/Cas9 mediated HDR (Homology Directed Repair) was utilized and in which the *PP2C* ORF was precisely replaced by the *hph* (hygromycin B phosphotransferase) ORF. Donor DNA consists of the *hph* ORF ligated to two 1 kb fragments flanking the target gene. The common plasmid vector pBluescript SK II⁺ (pBS-SK II⁺) as a carrier of the HDR template. Polyethyleneglycol-mediated protoplast transformation of *P. sojae* was described previously (Hua *et al.*, 2008).

Genomic DNA of potential resistant transformants were extracted and two rounds of primers were used in genomic PCR to check the HDR mutants (S1A Fig and S2 Table). The first round primers, F1/R1, (S1A Fig and S2 Table) were located inside the *PP2C* ORF to detect deletion events. As we can see in S1B Fig (F1/R1), the *PP2C* knockout mutants showed no

bands of *PP2C* sequence, indicating that deletion events may have occurred. Then, the second round of PCRs used primers located outside the *PP2C* homology arms (to avoid the interference of transferred plasmids) and within the *hph* gene to detect homologous recombination events. As we can see in S1B Fig (F2/R2, F2/R3), the *PP2C* knockout mutants showed clear bands of replaced sequences, indicating that the original *PP2C* ORF in the *P. sojae* genome had been replaced by the *hph* ORF. Furthermore, Sanger sequencing across the junctions of the flanking sequences and *hph* was also consistent with the replacement of the *PP2C* ORF with the *hph* gene (S1C Fig). For HK mutants, CRISPR/Cas9 mediated NHEJ (non-homologous end joining) was utilized, in which the direct deletions were generated in the *HK* ORF. To this end, more sgRNAs were included to guarantee the probability of cleavage in DNA fragments. The key region of this histidine kinase was located in its N-terminal, such as the TorS and Hpt domains. Thus four sgRNAs were centered around this key region (S1D and S1E Fig). The primer set F4/R4 (S1D and S1E Fig), located to cover this key region, was used to detect deletion events. As shown in S1F Fig, HK mutants exhibit clear smaller bands comparing to wild-type (P6497), indicating that direct deletion may have generated in *HK* ORF. For further verification of deleted sequence in HK sanger sequencing traces were introduced for track and visualization of the deletion regions. As displayed in S1D Fig, HK-Line1 has a deletion of 301bp and this deletion impairs Hpt domain of HK. HK-Line2 (S1E Fig) has a deletion of 2121bp and this deletion impairs both TorS and Hpt domain of HK.

## Mycelial growth and stress treatment

To determine the growth rate, 6-mm-diameter mycelia plugs were inoculated on V8 agar medium and incubated at 25°C in the dark. Sensitivity to stresses was evaluated on V8 agar medium, and on V8 agar medium supplemented with different concentrations of stress conditions. Colonial morphologies were photographed and colony diameters were measured over 4 days and used to calculate relative growth. Relative mycelial growth was shown on the diagram by stress-treated colony diameters with respective no-treated colony diameters (Mock). Data were subjected to statistical analysis by two tailed t-test. All assays were repeated at least three times.

## Production and morphology of sporangium

To quantify sporangia production, liquid 10% (vol/vol) V8 culture medium was inoculated with three round mycelial disks (7 mm in diameter) cut from a culture, and incubated for 48 h at 25°C in the dark. The mycelia were rinsed twice with sterile distilled water, and then flooded with sterile distilled water to stimulate sporangia formation. For counting of sporangia, after 8 h, sporulating hyphae was gently mixed in a blender to obtain a homogenous mixture. Subsequently, three random samples of 100 μL were taken and the number of sporangia in each sample was counted by microscopic examination at 40×magnification. All assays were repeated at least three times. Data were subjected to statistical analysis by two tailed t-test. All assays were repeated at least three times.

To observe sporangial morphology and cleavage, Congo red stain was added to the sporangia contained cultures 5 min prior to microscopic analysis. All assays were repeated at least three times.

## Zoospore chemotaxis and germination of cysts

Mycelia were rinsed five times with sterilized ddH$_2$O, about 3 mL of sterilized ddH$_2$O was added and the plates were incubated at 25°C for 8–10 h. The suspension was collected when zoospores were released in large quantities.

Root hairs of yellowing seedlings of HeFeng47 soybean were removed using forceps, placed in 500 μL of zoospore suspension, and observed under a microscope after standing for 5 minutes at 25˚C. Mobility was determined using at least two batches of independently obtained zoospores, and three independent experiments were performed.

Agarose plugs containing 15 μM daidzein were placed in microscopic chambers and photographs were taken after 20 min of incubation at room temperature. The chambers were filled with equal amounts of zoospore suspensions.

To quantify the germination of cysts, 500 μL of zoospore suspension was added to a 1.5mL EP tube, followed by liquid 10% V8 culture medium of equal volume. After 20 seconds of blending, the medium was placed in a 25˚C incubator for 3h, and 5μL of suspension was removed for assessment of the total numbers of resting and germinating cells under a Zeiss microscope. Data were subjected to statistical analysis by two tailed t-test. Asterisk indicates significant difference at *P<0.01*. The experiment was repeated three times, each in triplicate.

## Virulence and biomass assays

The soybean cultivar Hefeng 47, which is susceptible to *P. sojae* strain P6497, was grown in plastic pots containing vermiculite at 25˚C for 4 days in the dark. Zoospores were obtained as described previously and diluted to 100 zoospores/10 μL. Etiolated seedlings were inoculated by pipetting 10 μL of zoospore suspension onto hypocotyls, and were maintained in a climate-controlled room at 25˚C and 80% relative humidity in the dark. Symptoms were evaluated at 36-hour post-inoculation, and photographs were obtained. Three soybean hypocotyls inoculated by same *P. sojae* strain were collected as one sample. Genomic DNA was extracted using a plant DNA kit (Tiangen) from soybean samples for the biomass assay. Virulence was quantified by determining the ratio of *P. sojae* DNA (Primers: PsACTA-qRT-F/R, S2 Table) to soybean DNA (Primers: GmCYP2-qRT-F/R, S2 Table) in the infected plants by quantitative PCR. All assays were repeated independently at least three times.

To allow microscopic observation of the penetration of, and infectious hyphal expansion within soybean tissue, infected epidermal cells were collected at 12 and 24 hpi and soaked in Diaminobenzidine (DAB). After destaining in water, the infected epidermis cells were examined under a light microscope. Each strain was tested using at least two different preparations of zoospores, and five plants. All assays were repeated at least three times.

## Supporting information

**S1 Fig. Verification of mutants of PP2C, HK, and bZIP32.** (A) Schematic diagrams for knockout strategy of *PP2C*. CRISPR/Cas9 mediated HDR (Homology Directed Repair) was utilized for precisely replaced of the entire gene. Also, locations of the primers used to screen the PP2C knockout mutants (F1/R1, F2/R2, and F3/R3) are indicated. (B) Analysis of genomic DNA from the wild-type (WT) and PP2C-knockout mutants (ΔPP2C-Line1 and ΔPP2C-Line2) using the primers in (A) and actin primers (positive control). (C) Sanger sequencing traces of junction regions confirming that the PP2C was precisely replaced. Red dots, junction regions. (D and E) Schematic diagrams for deletion of key regions of HK from *P. sojae* genome. Four sgRNAs covering key regions of HK were designed and their locations were indicated by four red triangles. Sanger sequencing traces around the deletion regions in ΔHK-Line1 (D) and ΔHK-Line2 (E). The wild-type sequence was represented above the mutated ones, with PAM (Protospacer Adjacent Motif) was indicated with red and cleavage sites was indicated with red arrows. (F) Genomic analysis of the deletion events in HK-knockout mutants (ΔHK-Line1 and ΔHK-Line2), comparing with the wild-type (WT). Primers (F4/R4) were used to covering the key region of HK. Actin gene was included as positive control.

(G) The expression analysis of *bZIP32* in bZIP32-Line1 and bZIP32-Line2. The bar (SD) represents the relative expression level calculated by quantitative reverse transcription PCR using the $2^{-\Delta\Delta Ct}$ method. The level of gene expression in the wild-type was set equal to 1 and used to calculate the relative expression levels of the genes in the transformants. All experiments were repeated three times with similar results. **Significant difference at *p < 0.01*. (TIF)

**S2 Fig. Phenotypes of HK and bZIP32 mutants at the stages of mycelial and sporangium stages.** (A) Growth characteristics of the wild-type (P6497), HK and bZIP32 mutants (PP2C-Line1 and PP2C-Line2, bZIP32-Line1 and bZIP32-Line2) on V8 agar medium without (Mock) or with treatments of NaCl, Sorbitol or $H_2O_2$. (B) Relative mycelial growth of wild-type (P6497), HK and bZIP32 mutants on V8 agar medium (Mock). (C) Relative colony diameters of wild-type (P6497), HK and bZIP32 mutants after 4 days of growth in different stresses. And Relative colony diameters were calculated for each treatment relative to growth on V8 agar medium only. (D) Upper: The numbers of sporangia in wild-type (P6497), HK and bZIP32 mutants. The sporangia morphology (middle) and cytoplasm cleavage (bottom) within sporangia were observed in the wild-type (P6497), HK and bZIP32 mutants. (E) The relative numbers of sporangia in the wild-type (P6497), HK and bZIP32 mutants. (TIF)

**S1 Table. Information for the genes in clustering analysis displayed in Fig 2.** (XLSX)

**S2 Table. Primers used in this study.** (XLSX)

**S3 Table. Domain annotations of proteins included in Figs 3A, 4A and 5A.** (XLSX)

## Acknowledgments

We would like to thank Dr. Yufeng Fang and Dr. Brett Tyler (Oregon State University, Corvallis) for the technical assistance. We thank Dr. Suomeng Dong, Dr. Yan Wang, and Kaixuan Duan (Nanjing Agricultural University, Nanjing) for helpful discussions. Bioinformatics analyses were supported by the high-performance computing platform of Bioinformatics Center, Nanjing Agricultural University.

## Author Contributions

**Conceptualization:** Min Qiu, Yuanchao Wang.

**Data curation:** Wenwu Ye, Ming Wang.

**Formal analysis:** Mengjun Tian.

**Funding acquisition:** Min Qiu, Ming Wang, Yuanchao Wang.

**Investigation:** Min Qiu, Saijiang Yong, Yaru Sun, Jingting Cao, Yaning Li.

**Methodology:** Min Qiu, Mengjun Tian, Ming Wang.

**Project administration:** Min Qiu.

**Resources:** Xin Zhang, Chunhua Zhai, Wenwu Ye.

**Supervision:** Ming Wang, Yuanchao Wang.

**Validation:** Min Qiu, Mengjun Tian.

**Visualization:** Min Qiu, Mengjun Tian.

**Writing – original draft:** Min Qiu.

**Writing – review & editing:** Min Qiu, Mengjun Tian, Ming Wang, Yuanchao Wang.

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
