## [Decision Letter · Decision Letter 0]

16 Nov 2022

Dear Dr. Wang,

Thank you very much for submitting your manuscript "Phase-specific transcriptional patterns of the oomycete pathogen Phytophthora sojae unravel genes essential for asexual development and pathogenic processes" for consideration at PLOS Pathogens. As with all papers reviewed by the journal, your manuscript was reviewed by members of the editorial board and by several independent reviewers. Whilst the reviewers were all interested in the work they have raised major concerns that preclude publication in its current form. In light of the reviews (below this email), we would like to invite the resubmission of a significantly-revised version that takes into account the reviewers' comments.

We cannot make any decision about publication until we have seen the revised manuscript and your response to the reviewers' comments. Your revised manuscript is also likely to be sent to reviewers for further evaluation.

Sincerely,

Paul Birch

Academic Editor

PLOS Pathogens

Bart Thomma

Section Editor

PLOS Pathogens

Kasturi Haldar

Editor-in-Chief

PLOS Pathogens

orcid.org/0000-0001-5065-158X

Michael Malim

Editor-in-Chief

PLOS Pathogens

orcid.org/0000-0002-7699-2064

Reviewer's Responses to Questions

**Part I - Summary**

Reviewer #1: Let me start by affirming that this is a very useful clustering analysis that others can use in gene discovery experiments. I very much like the organization and presentation of the figures.

However the Gene ID numbers cited is this paper are recognizable as ones associated with earlier databases, and do not correspond (in my tests ) to the gene id system of P sojae in Fungidb, where all genes are identified as Physodraft_123456. The authors need to add a row to identify equivalent genes in this database for all of their supplementary tables.

Other labs may want to use the information in Fig. 2 to characterize specific genes Therefore the Gene id information for all the clusters in Fig.2 must also be included as a supplementary table

Make a supplementary document which enables the reader to easily discern the meaning of the supplementary tables

Reviewer #2: Aimed at identifying novel functional genes for precisely controlling oomycete diseases, this research investigated infection and developmental stage-specific gene expression in the oomycete pathogen Phytophthora sojae, using available transcriptome data. By performing hierarchical clustering, specification, and diversification analyses, the authors focused on the developmental stages that are more remarkable in transcriptional plasticity, leading to the genes being resolved into six distinct modules. The authors took good advantages of genome editing in P. sojae and generated knockout mutants for three candidate stage-specific genes.

The research generated lot of information useful for understand P. sojae development. However, the novelty and in-depth data analysis are not sufficient, in neither gene expression nor knockout mutants.

Reviewer #3: Phase-specific transcriptional patterns of the oomycete pathogen Phytophthora sojae unravel genes essential for asexual development and pathogenic processes” by Qiu et al. is an interesting work. I have read this paper with great interest and but found it not so easy to follow. I believe the manuscript could be further improved from the suggested changes in the attached document.

**Part II – Major Issues: Key Experiments Required for Acceptance**

Reviewer #1: no additional experiments are needed

Reviewer #2: 1. Data compilation is significant, and too many speculations in expression data analyses.

2. The GO enrichment analysis described in Fig2 needs to be simplified.

3. The genes up-regulated in the infection stage were mentioned many times in the manuscript, but the control was not explained.

4. In Fig3B, when supplemented with different concentrations of NaCl, sorbitol or H2O2, the colony is smaller than that of the mock. The colony of the two mutants shown in Fig3C is different after supplemented with sorbitol. In a word, the information shown in the pictures is inconsistent with the description in the manuscript. Furthermore, what is the purpose of this experiment?

5. The reasons for selecting the three genes representing different development stages need to be explained more fully. Since genes of different development stages were focused as stated, why selected PP2C, HK and bZIP32 that were up-regulated during infection? Furthermore, knockout mutants of these genes were not examined for their roles in plant infection, given that they were up-regulated during infection. In addition, high expression or up-regulated expression need more detailed description.

6. Silencing efficiencies of target genes in all mutants are not provided.

7. According to the description in the manuscript, the genes in M4 and M6 were highly expressed not only in mycelia and sporangia stages, but also in the other three development stages. The author finally took the genes encoding serine/threonine phosphatases in M4 as candidates and selected PP2C as the target gene. Although PP2C was proved to affect colony morphology and the number of sporangia, how is its role in other development stages?

8. HK-line2 is not available in Fig6A, statistical analysis on lesion area is also needed.

9. How is the phenotype change of HK and bZIP32 mutants in the development stages of mycelia and sporangia.

10. The model in Figure7 is poorly proposed, a simple model is suggested.

Reviewer #3: Please see the attached document.

**Part III – Minor Issues: Editorial and Data Presentation Modifications**

Reviewer #1: The use of the word “meanwhile” by the authors is in all cases inappropriate

Abstract

This is awkwardly phrased

Therefore, this study reboots the analysis from available transcriptome data

in a model oomycete pathogen, Phytophthora sojae, to excavate more functional

genes. A set of expression matrix of 10,953 genes across ten stages (including

development and infection) were applied to recognize the transcriptional changes

underlying critical stages for disease spreading.

Awkard

Unlike plant diseases caused by filamentous fungi, most oomycete diseases spread

60 quickly in the fields, relying on their energetic zoospores for better colonization through

61 chemotaxis

Try

The motile zoospore stage facilitate the dispersal of these pathogens under moist or wet conditions, and(Maltese et al., 1995, Judelson & Blanco, 2005). the zoospores are able to use both general (ethanol) and host-specific signals to swim towards host roots (Maltese et al., 1995, Judelson & Blanco, 2005).

Check:

Most oomycetes produce

62 numerous multinucleate sporangia at the end of specialized hyphae within the diseased

63 hosts (Maltese et al., 1995).

I am assuming the authors mean on the surface of plant tissues…..

.

Despite such advances in oomycetes kingdom, there are still

90 many unknowns on molecular mechanisms underlying their life cycles, especially for

me critical stages in spreading of oomycete diseases.

Here, we describe the employed large-scale transcriptome profiling to analyze the dynamics

98 of the P. sojae life cycle.the large-scale transcriptome profiling to analyze the dynamics

98 of the P. sojae life cycle.

147 during the developmental stages and possessed a highly dynamic transcriptome, which

148 referred as we describe as transcriptional plasticity.

Awkwardly phrased

To better recognize the gene expression patterns in the P. sojae life cycle, considering

152 the greater transcriptional plasticity, transcript matrix of five developmental stages were

153 further analyzed.

Perhaps…..

Next we utilized the pattern of greater transcriptional plasticity across the five developmental stages to identify novel gene expression patterns

Re

L158 As a result, six modules consisting of 7029 genes were obtained on the basis of expression clustering, which were portrayed in the radar maps in Fig. 2.

I feel this is too much like “just go look at the figure “without giving the reader a visual description of the data analysis strategy. A much better job is done in the figure legend and some of this information must be replicated in the text.

Follow-up sentence(s) enabling the reader to see what was accomplished with this anlysis is clearly needed.

Change:

169 Meanwhile, The other modules, M1 and M5, with 1553 and 605 genes, respectively…..

193 Meanwhile, t The fourth module (M4) was the largest among these three

193 modules, with 2391 genes; while M3 had 1539 genes; and M6 only 501 gene

Lines 204 to 215

We noted that the M4 and M6 modules contained several genes in the serine/threonine phosphatase and serine/threonine kinase families and elected to investigate the expression patterns of these developmental regulators further. In some cases, genes associated with the same function exhibited differences

208 across stages in the mRNA abundance levels, probably due to the differences in the

209 importance of these genes in molecular functions. Then, to To address this issue, we

210 performed statistical clustering of mRNAs from all stages and visual inspection of

211 protein domains to categorize the genes into subclasses. We classified the genes from

212 the serine/threonine phosphatase cluster into two groups, one that was relatively less

213 expressed and the other that was upregulated at all stages of infection (Fig. 3A).

We also identified multiple phosphatase domains, such as PP2A, CDC55, PP2C, HP, and

215 DSP, in serine/threonine phosphatase-encoding genes.

Meanwhile, IN contrast zoospores of the HK knockout mutants (HK-Line1 and HK-Line2)

Were not chemoattracted to root hairs. Zoospores of the wild-type strain formed cysts on and adjacent to this regionof the root. While only a few cysts were observed in this region of the root in the knockout mutant strains.

(Fig. 4B, Root Hair).

Change

Based on the above results, the following model can be proposed (Fig. 7). We

380 described the large-scale profiling of the P. sojae transcriptional changes during the

381 transition between developmental stages. Meanwhile, we identified several candidate

382 genes among the various stages of P. sojae life cycle, with significant serine/threonine

383 phosphatase activity at the mycelial and sporangium stages, phosphotransferase activity

384 at the zoospore and cyst stages, and transcription activity during cyst germination (Fig.

385 7). Subsequently, by taking advantage of genome editing techniques, the absence of

386 these candidate genes verified their role during specific stage transitions, as well as

387 pathogenic processes (Fig. 7).

In summary analysis of transcriptional profiling across the different developmental stages of P sojae. Has enabled us to functionally characterize several key developmental genes facilitating the shift of gene expression form one stage to the next. (fig. 7). These include. …..

Awkwardly phrased ..

447 2019). We found that, similar to fungi, P. sojae has a hybrid HK, this protein were

448 integrated HK with RR domains (Fig. 4A) and involved in zoospore chemotaxis

449 towards hosts and the subsequent cyst germination process (Fig. 4B).

Reviewer #2: a. Fig S1 is not marked with E.

b. Line138-139: SP and MY should be marked.

c. Line189: M8?

d. Line282: he?

Reviewer #3: Please see the attached document.

PLOS authors have the option to publish the peer review history of their article (what does this mean?). If published, this will include your full peer review and any attached files.

Reviewer #1: No

Reviewer #2: No

Reviewer #3: No
---

## [Decision Letter · Decision Letter 1]

28 Feb 2023

Dear Dr. Wang,

We are pleased to inform you that your manuscript 'Phase-specific transcriptional patterns of the oomycete pathogen Phytophthora sojae unravel genes essential for asexual development and pathogenic processes' has been provisionally accepted for publication in PLOS Pathogens.

Best regards,

Paul Birch

Academic Editor

PLOS Pathogens

Bart Thomma

Section Editor

PLOS Pathogens

Kasturi Haldar

Editor-in-Chief

PLOS Pathogens

orcid.org/0000-0001-5065-158X

Michael Malim

Editor-in-Chief

PLOS Pathogens

orcid.org/0000-0002-7699-2064

Reviewer Comments (if any, and for reference):

Reviewer's Responses to Questions

**Part I - Summary**

Reviewer #1: I am satisfied with this version of the manuscript

Reviewer #2: The manuscript has been significantly improved

Reviewer #3: The authors have addressed all my comments and i have no further comments.

**Part II – Major Issues: Key Experiments Required for Acceptance**

Reviewer #1: none

Reviewer #2: (No Response)

Reviewer #3: (No Response)

**Part III – Minor Issues: Editorial and Data Presentation Modifications**

Reviewer #1: no further issues

Reviewer #2: (No Response)

Reviewer #3: (No Response)

PLOS authors have the option to publish the peer review history of their article (what does this mean?). If published, this will include your full peer review and any attached files.

Reviewer #1: **Yes: **Paul F. Morris

Reviewer #2: No

Reviewer #3: No

---

## [Editor Report · Acceptance letter]

20 Mar 2023

Dear Dr. Wang,

We are delighted to inform you that your manuscript, "Phase-specific transcriptional patterns of the oomycete pathogen Phytophthora sojae unravel genes essential for asexual development and pathogenic processes," has been formally accepted for publication in PLOS Pathogens.

Best regards,

Kasturi Haldar

Editor-in-Chief

PLOS Pathogens

orcid.org/0000-0001-5065-158X

Michael Malim

Editor-in-Chief

PLOS Pathogens

orcid.org/0000-0002-7699-2064